# On the relationship between atmospheric rivers, weather types and floods in Galicia (NW Spain)

Jorge Eiras-Barca[1], Nieves Lorenzo[2], Juan Taboada[3], Alba Robles[2], and Gonzalo Miguez-Macho[1]

[1]Non-Linear Physics Group, Universidade de Santiago de Compostela, Galicia, Spain.
[2]Ephyslab, Universidade de Vigo, Galicia, Spain.
[3]MeteoGalicia, Consellere Medio Ambiente, Xunta de Galicia, Spain.

*Correspondence to:* jorge.eiras@usc.es

**Abstract.**

Atmospheric rivers (ARs) –long and narrow structures of anomalously high water vapor flux located in the warm sector of extratropical cyclones– have been shown to be closely related to extreme precipitation and flooding. In this paper we analyze the connection between ARs and ~~floods~~flooding in the northwestern Spanish region of Galicia under ~~different~~a variety of synoptic conditions represented by the so-called "weather types", a classification of daily sea level pressure patterns obtained by means of a simple scheme that adopts the subjective procedure of Lamb. Flood events are identified from official reports ~~of the~~conducted by the Spanish Emergency Agency from 1979 to 2010. Our results suggest that although most flood events in Galicia do not coincide with the presence of an overhead AR, the latter are present in the majority of severe cases, particularly in coastal areas. Flood events associated with ARs are connected to cyclonic weather types with westerly and southwesterly flows and which occur mostly in winter months. The link between ARs and severe flooding is not very apparent in inland areas or during summer months, in which cases heavy precipitation is usually not frontal in nature ~~but is~~but rather, convective. Nevertheless, our results show that, in general, the amount of precipitation in flood events in Galicia more than doubles when an AR is present.

## 1 Introduction

Atmospheric rivers (ARs) are narrow, elongated structures that carry high quantities of water vapor in the lower troposphere. The climatological characteristics of ARs have been recently reviewed by Guan and Waliser (2015), who ~~showed~~proved that they exhibit a mean length of about 3,700 km, and have average integrated vapor transport (IVT) fields of $375 \, \text{kg} \cdot \text{m}^{-1} \cdot \text{s}^{-1}$. ARs are usually found in the warm sector of extratropical cyclones, and are associated with the meridional transport of latent and sensible heat from the (sub)tropics to mid-latitudes (Newell et al., 1992; Zhu and Newell, 1998; Gimeno et al., 2010; Ralph and Dettinger, 2011; Lavers and Villarini, 2013; Matrosov, 2013; Neiman et al., 2013; Rutz et al., 2014; Gimeno et al., 2014; Garaboa-Paz et al., 2015; Gimeno et al., 2016; Waliser and Guan, 2017).

There has been a rise in the development of detection algorithms ~~of~~for ARs over the past few years, with significant contributions made by numerous authors (e.g. Lavers et al., 2012; Guan and Waliser, 2015; Eiras-Barca et al., 2016; Brands et al., 2016). Despite the fact that discrepancies in the finer details of their detection remain, all algorithms in the literature rely on an anal-

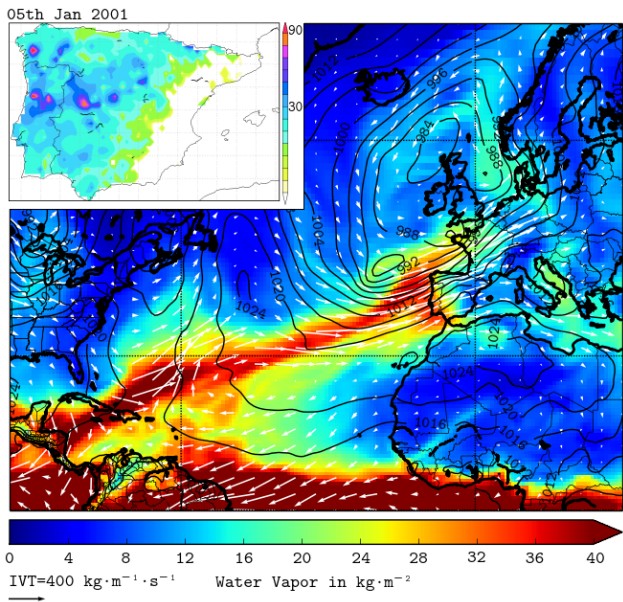

**Figure 1.** An AR as it makes landfall in Galicia (NW Spain) on January 5th of 2001. The field represents the IWV ($\mathrm{kg \cdot m^{-2}}$) while the arrows symbolize the integrated water vapor flux or IVT ($\mathrm{kg \cdot m^{-1} \cdot s^{-1}}$). The upper left hand image displays extreme precipitation (more than 90 mm in 24 h) accumulated throughout the Iberian Peninsula.

ysis of the integrated water vapor column (IWV) and IVT fields. ARs are always characterized by highly ~~enhanced~~increased values of both variables when compared with surrounding areas. An example of a well-defined AR ~~landfalling~~making landfall on the Iberian Peninsula coast, ~~together~~along with the associated extreme precipitation, can be found in Figure 1.

~~A major natural disaster that humans are faced with today are flood events (FEs), where extensive socioeconomic impacts and~~
~~fatalities are usually associated with flooding episodes worldwide.~~ Flood events (FEs) are major natural disasters that humans face today, as flooding episodes worldwide are commonly associated with extensive socioeconomic impacts and fatalities. The correlation and causality between ARs and extreme precipitation events has been soundly demonstrated in many parts of the world, including the Iberian Peninsula (Ramos et al., 2015; Eiras-Barca et al., 2016). Nevertheless, few studies have used flood databases to analyze how the impact of ARs is reflected in people's daily lives. In this regard, the connection between ARs
and floods has been extensively ~~shown~~established for the west coast of the United States of America (e.g. Ralph et al., 2006; Dettinger, 2011), as well as for some European regions (e.g. Lavers et al., 2011).

The region of Galicia is located in the northwestern part of Spain. The Galician climate is highly influenced by its location within the North Atlantic storm track, where a continuous passage of baroclinic structures increases the possibility of heavy rain episodes (Nieto et al., 2011). This is especially true in winter and autumn, whereas in spring and summer, as the storm
track moves poleward, intense precipitation associated with convective episodes plays a more prominent role (Eiras-Barca et al., 2016).

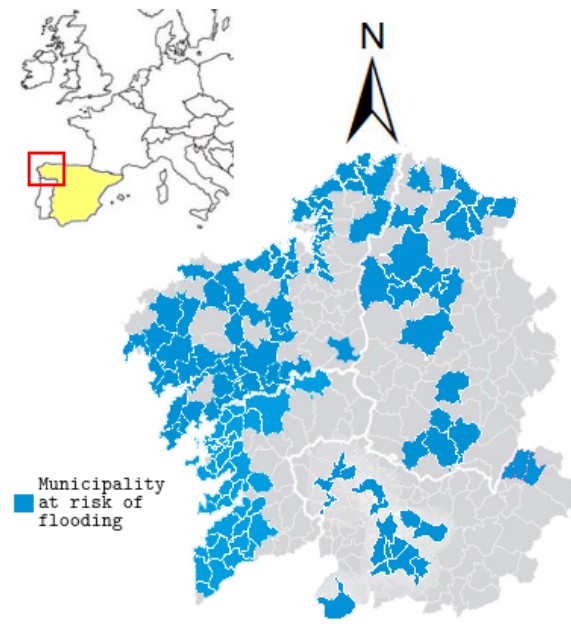

**Figure 2.** Municipalities of Galicia that are at regular risk of flooding. Source: Regional Government and Miño-Sil river authority.

The interannual variability of rainfall in Galicia is mostly linked to certain modes of the atmosphere (e.g. Lorenzo and Taboada, 2005; Lorenzo et al., 2006, 2008, 2011), ~~in particular~~particularly the North Atlantic Oscillation (NAO), which modulates the position of the storm track and its impact in this region. Therefore, higher correlations that explain the variability in precipitation, especially in southern Galicia, correspond to this index (Trigo et al., 2008). Notwithstanding, other teleconnection patterns such as the Eastern Atlantic (EA) or the Scandinavian Pattern (SCAND) should also be taken into account when explaining the ~~inter-annual~~interannual variability of precipitation and ~~AR activity~~AR activity in Galicia (e.g. Lorenzo et al., 2006; Bueh and Nakamura, 2007; Ramos et al., 2015).

Extreme precipitation and anomalous winds are the most frequent climate ~~risk~~risks in Galicia. In coastal towns, floods are more common (Figure 2) and the corresponding socioeconomic impacts are the most severe (Martínez and Ezpeleta, 2000). Fatalities and serious damage to transportation and communication systems are common when FEs occur. ~~Because of this~~Due to such, a better understanding of the meteorological conditions that produce FEs is crucial for the people and economy of this Atlantic region (Cabalar-Fuentes, 2005).

One way to integrate different meteorological parameters, such as rainfall, direction and wind intensity, or temperature, into a single index is via the classification of synoptic situations (i.e. weather types, WTs). Using data such as the sea level pressure (SLP) or geopotential height for classification, each ~~considered~~observed time period is assigned a WT, ~~which allows~~allowing one to study the associated meteorological variability and its consequences in different fields in a simple and ~~easy-to-interpret way~~decipherable way. In this ~~paper~~article, we ~~will~~ look at the occurrence of FEs and their related WTs in Galicia.

**Table 1.** Total number of FEs registered in Galicia throughout the period 1979-2010

| Season | SIL Region | COSTA Region |
|--------|-----------|--------------|
| DJF | 206 | 190 |
| MAM | 45 | 102 |
| JJA | 9 | 102 |
| SON | 97 | 154 |

~~In this study we will~~With the following reaserch, we analyze the connection between ARs and FEs under different WT synoptic situations. The scope of this paper is twofold. ~~First, we~~We first identify the relationship between ARs and FEs under different synoptic conditions in the studied region. ~~Next, we show that this study may be useful to properly understand and predict FEs.~~We then demonstrate how this study may be of use in properly understanding and predicting FEs. The structure of the paper is as follows: In Section 2 we present our analysis methods, while in Section 3 we outline our obtained results~~,~~ and give a brief analysis. Finally, our conclusions are presented in Section 4.

## 2 Data and Methods

This research aims to analyze the connection between ARs and flood events. Therefore, rather than extrapolate ordinary precipitation databases, we have employed a flooding events database published by the Spanish Emergency System (Protección Civil de España), where only occurences with serious implications, in terms of damage, are considered.

FEs were gathered from official reports published by the Spanish Emergency Service. This database registers FEs over an extended period, i.e. from 1979 to 2010 (Interior, 2014). Two different areas were ~~separately~~individually analyzed: the Galician Coast (COSTA) and the Miño-Sil (SIL) hydrological units (river basins). Whereas the former unit encompasses all smaller Atlantic basins and is representative of coastal towns, the latter corresponds to the Miño-Sil river basin, and depicts conditions inland of Galicia. The rainfall database has been used to quantify the exact amount of precipitation only on flooding days.

A total number of 754 AR events have been detected ~~for~~on the Galician coast throughout ~~the period~~ years 1979-2010. In the same period, 357 and 548 flood events were registered for the SIL and COSTA regions respectively. The seasonal distribution of FEs is ~~given~~provided in Table 1.

The database published by Guan and Waliser (2015) was used in the detection of ARs. This is an advanced AR database, ~~which is able~~with the ability to identify ARs ~~by~~using complex considerations regarding the characteristics of IWV and IVT fields in terms of coherence and continuity (Waliser and Guan, 2017). Equations 1 and 2 represent the methodology for the

integration of the IWV and IVT fields, respectively, where $q$ is the specific humidity, $g$ is the gravitational force, $\boldsymbol{u}$ is the horizontal wind field and the integration covers the whole troposphere from the first pressure level ($P_0$) to the top ($P_f$).

$$IWV = \frac{1}{g}\int_{P_0}^{P_f} q \cdot dp \tag{1}$$

$$IVT = |\frac{1}{g}\int_{P_0}^{P_f} q \cdot \boldsymbol{u} \cdot dp| \tag{2}$$

~~Only days where an AR landfalls onto the Galician coast (from 41.5°N to 44°N at 9°W) have been taken into consideration in this analysis.~~ Only days in which ARs made landfall onto the Galician coast have been taken into consideration throughout this analysis like AR-days. The classification of synoptic situations was done by adopting the procedure developed by Trigo et al. (2000b), which was adapted from Jenkinson and Collison (1977) and Jones et al. (1993). The southerly flow (SF), westerly flow (WF), total flow (TF), southerly shear vorticity (ZS), westerly shear vorticity (ZW) and total shear vorticity (Z) were

computed using sea level pressure (SLP) values collected for the 16 grid points shown in the ~~supplementary material~~ Figure A1 (Lorenzo et al., 2008). For the index calculations, we applied the equations outlined by Lorenzo et al. (2008) to an NCAR reanalysis of size $2.5° \times 2.5°$. We used the conditions established by Trigo et al. (2000b) to define the different WTs. For the sake of simplicity, only WTs that appeared with a frequency of over 3% are considered in this study. Under this condition, the total number of WTs is 9 ~~in~~throughout the extended winter months (ONDJFM), and 12 ~~in~~throughout the extended summer

15 months (AMJJAS). SLP composites for each of the considered WTs are shown in Figure 3. A brief description of each WT can be found in Table 2.The hybrid WT are a combination of both WTs. Finally, ECMWF Era-Interim reanalysis data (Dee et al., 2011) were used to make composites of the SLP, IVT and IWV variables on days with or without the occurrence of ARs and FEs.

## 3   Results and discussion

The connection between WTs and FEs has been studied ~~via~~using two ~~different~~distinct procedures. First, in Section 3.1 we refer to the aforementioned connection in terms of the corresponding WTs that they are associated with ~~it~~. Then in Section 3.2, we demonstrate that the anomaly composites for different variables show whether or not an AR is detected ~~together~~simultaneously with ~~a~~an FE. Finally, the set of meteorological stations stated in Table A1 is used to quantify the amount of precipitation that occurred throughout the FEs.

### 3.1   Weather Types

Figure 4 shows the frequency of occurrence for each of the winter and summer WTs during the FEs, for both (COSTA and SIL) regions. For these two regions, ARs are primarily correlated with FEs in winter months. FEs during summer months are

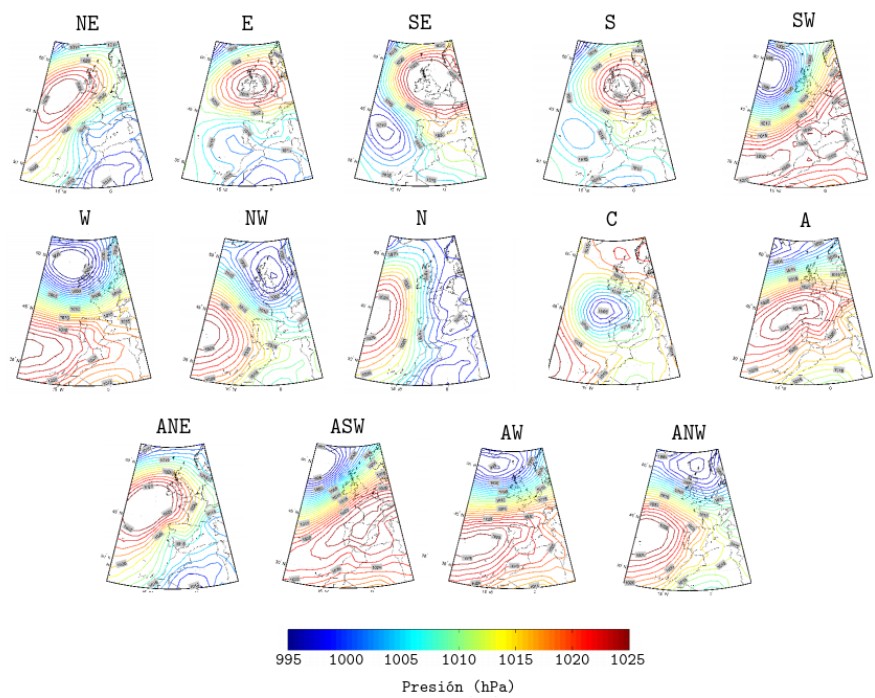

**Figure 3.** SLP composites for all the WTs considered in this analysis.

**Table 2.** A brief description of the pure predominant WTs used in the analysis. The bulk of this information was obtained by Ramos (2012).

| WT | Season | Brief description |
|---|---|---|
| NE | Sum | Extended high-pressure settled over the west of Ireland and low-pressure in the Mediterranean Sea. |
| E | Sum | Synoptic situations characterized by high-pressure over the British Isles and low-pressure dominating in North Africa. |
| SE | Win | Low-pressure extending towards Madeira and high-pressure over Northern Europe. |
| S | Sum,Win | High-pressure over the British Isles and low-pressure established in the North Atlantic (Azores region). |
| SW | Win | Low-pressure system to the west of Ireland with a large anticyclone over the Mediterranean region. |
| W | Sum,Win | Low-pressure system over the North Atlantic with a high-pressure system over the Azores. |
| NW | Sum,Win | Low-pressure system over the British Islands and a anticyclone system located over Azores. |
| N | Sum | Azores high-pressure near the Islands and a low-pressure over southern Europe and the Mediterranean basin. |
| C | Sum,Win | Low-pressure centre over the NW Iberian Peninsula |
| A | Sum,Win | Extended high-pressure centre between the Iberian Peninsula and the Azores Islands. |

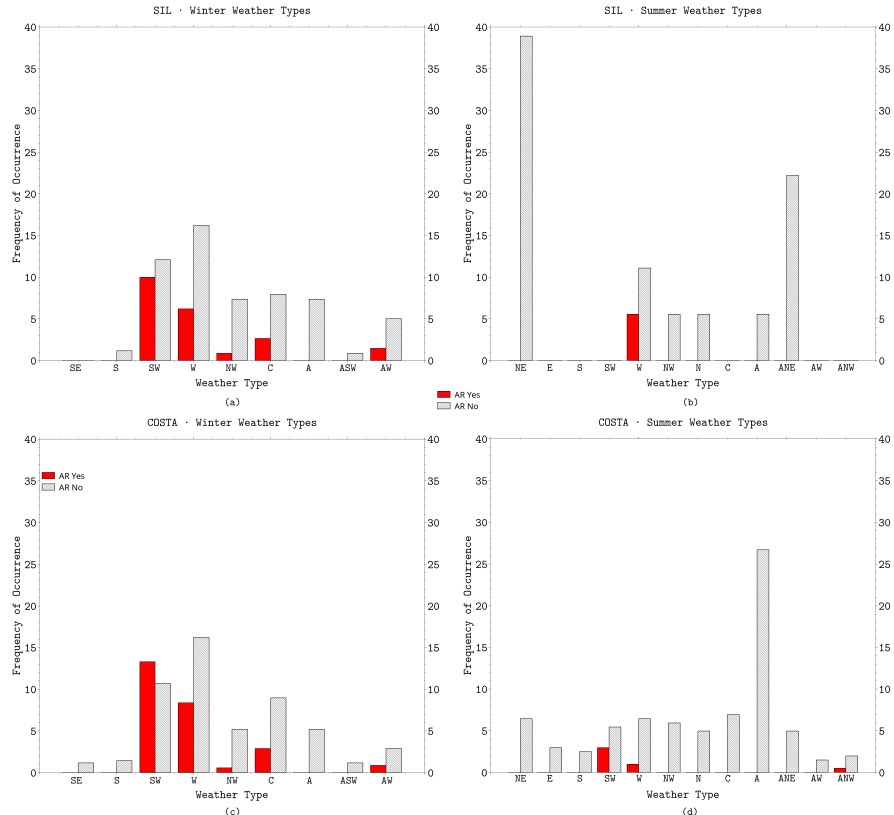

**Figure 4.** Frequency of occurrence for each WT with FE for the SIL region in the extended winter (ONDJFM), a) and extended summer (AMJJAS), b) months, as well as for the COSTA region in the extended winter (c) and extended summer (d) months. Red bars represent WTs associated with FEs when an AR has been simultaneously detected, while black lined bars refer to FEs that do not have an associated AR event.

mostly associated with anticyclonic situations (NE, A, AE) that block the arrival of fronts, and ~~thus~~ ARs. This is in line with the convective character of extreme precipitation in the extended summer months for these regions. In winter months, however, FEs are mostly associated with W, SW, C, and NW WTs, all ~~of them~~of which are connected with cyclonic synoptic situations or situations with humid west flux. Under these conditions, the arrival of fronts to the Galician coast is very likely. ~~Especially~~

5  ~~remarkable is the case of the SW WT, which~~The case of the SW weather type is particularly remarkable, as it registers as many floods associated with AR events as floods not associated with AR events. As Figure 3 indicates, this WT is characterized by the presence of a cyclone near the western British Isles, which enhances the arrival of fronts to the Iberian coast. The same applies, though to a lesser extent, ~~with~~to the W, C and NW types.

However, it is necessary to consider that periods of flooding do not always coincide with periods of extreme rainfall, ~~since~~due

10  to the fact that one or two days of heavy rainfall can produce floods that can be maintained over time simply with rainfall quantities that are close to normal amounts. Therefore, to study in more detail the relationship between FEs and the presence

of ARs, we have chosen the day with the highest rainfall in each flood period and looked for the presence of ARs on that specific date. Most FEs did not simultaneously occur with an AR when all days of an FE were considered. However, Table A2, exhibiting days where maximum precipitation overlaps with AR detection, indicates that for the COSTA basin, 16 out of the 23 days with the highest rainfall in the FEs (70%) took place with the presence of an AR over Galicia. In the SIL region (Table A3), 7 out of 17 (41%) coexisted simultaneously with an AR. However, if only the extended winter is analyzed in the same database, ratios of 15 out of 19 (79%) and 7 out of 13 (54%) appear for the COSTA and SIL basins, respectively. Once again, our results point to the key role that ARs play in FEs in the COSTA region. This role is diminished ~~in the inner points~~farther inland, due to the more likely convective character of extreme precipitation over ~~them~~these areas.

Figure 5 shows, for each WT and each region, the precipitation ratio occurring when an AR is (or is not) detected, in the context of all the precipitation that occurred in the corresponding WT when there was flooding.

First, the mean precipitation was calculated for each WT, then we computed the accumulated precipitation during the flooding days simultaneously with the WT under consideration. ~~Then~~Subsequently, the mean precipitation in FEs was calculated and separately considered according to the AR detection. The cited ratio is the result of the division of the latter by the former. Thus, this ratio quantifies how much precipitation should be expected when an FE coincides with an AR, with respect to ~~a~~an FE that is not accompanied by an AR for one of the WTs considered. In general, the expected rainfall on a day with an AR within a flood period is more than double ~~that~~than that which is expected on the same day without an AR. Especially noteworthy is the case of the ANW type for the SIL and COSTA regions, and the NW for the SIL region, where the expected amount of winter rainfall was five times larger than if the FE coincided with an AR (relative to when no AR was detected).

The same occurs, although to a lesser extent, for types SW, W and C for winter precipitation in SIL, and with types SW, W, NW and C for winter precipitation in COSTA. In the summer months, the same occurs for the W types for the interior region (SIL) and the AN type for coastal regions. Even when SW is not the predominant weather type in the summer months, the occurrence of ~~this~~the WT in this season represents the few fronts with ARs arriving on the Galician coast. It is uncommon to observe AR precipitation from WTs other than W or SW in summer months.

Previous works that analyzed precipitation using the methodology of WTs concluded that most of the yearly and winter precipitation is associated with WTs C, SW and W (e.g. Trigo et al., 2000a; Lorenzo et al., 2008; Cortesi et al., 2014). Our results ~~are in complete agreement~~corroborated with those obtained by the cited studies, ~~adding~~contributing the idea that ARs are responsible for most of this precipitation.

## 3.2 Anomaly Maps

Figure 6 shows anomaly composites with regard to the mean sea level pressure for each point in the Atlantic Ocean, which delineates when an AR is and is not detected in Galicia, as well as when an FE is and is not detected over the same region for both the extended winter and summer months. With ~~regard~~regards to the winter maps, and always speaking in terms of the most probable situation, for an AR to be detected on the coast of Galicia, there would have to be a convergence of a high-pressure center to the south of it, and a low-pressure center to the north. For an AR to be detected with ~~a~~an FE, the previously described situation would have to occur, and the low-pressure center would have to occupy a very large space over the North Atlantic.

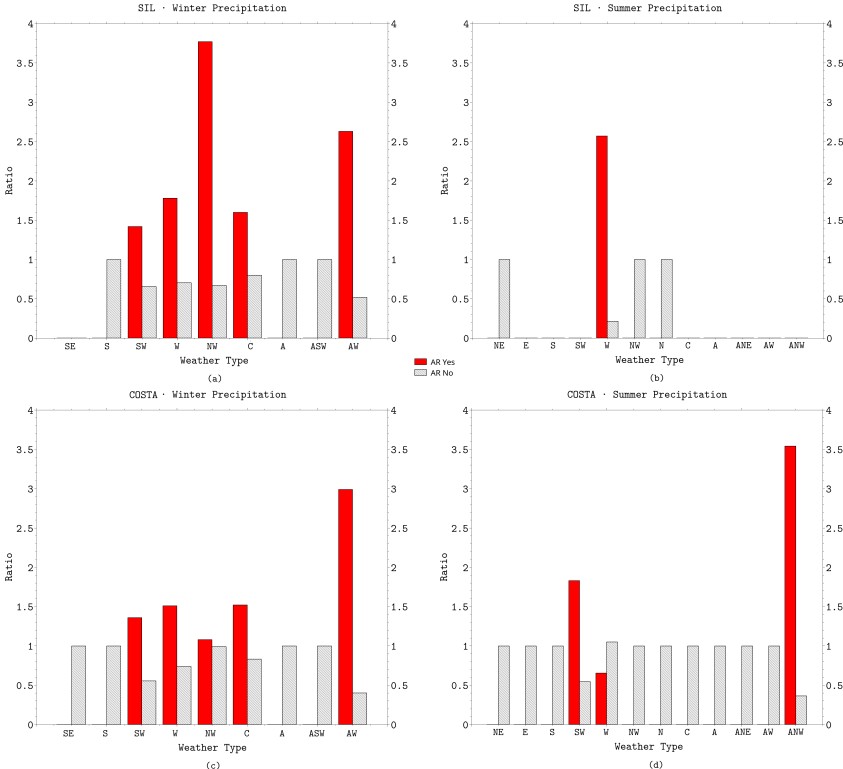

**Figure 5.** Same as Figure 4 but with precipitation ratio.

Flooding alone, with no AR presence, would occur with any similar situation to that described above, but both pressure centers would be weakened. On the days ~~that~~in which neither ARs nor floods were detected, a well-developed anticyclone over the Azores was identified as being influential in blocking the passage of baroclinic systems.

In the case of the summer months, the situation is similar to that of the ~~the~~ winter months as far as AR detection is concerned, except with FEs that occurred concurrently with a well-developed anticyclone over the Azores islands. This peculiarity shows that floods in the summer months are not as closely related to the arrival of baroclinic structures as they are to convective precipitation, which is compatible with the idea that the Azores anticyclone forms a blocking event.

Figure 7 is analogous to Figure 6, but shows anomalies in the IVT fields instead of the SLP. The results indicate that the detection of an AR in the winter months is contingent ~~on~~upon the presence of an intense IVT field over Galicia. These results are intensified when the AR is accompanied by ~~a~~an FE. In cases where no AR is detected, no intense IVT fields are observed over the ~~study~~studied region, especially when an FE is also not detected. With respect to the summer months, no significant qualitative differences are detected in relation to the winter months.

Figure A2 is comparable to Figures 6 and 7, but herein the IWV field is represented. ~~With respect to this figure~~As revealed by this figure, both for the summer and winter months, more pronounced IWV anomalies were observed for Galicia when an

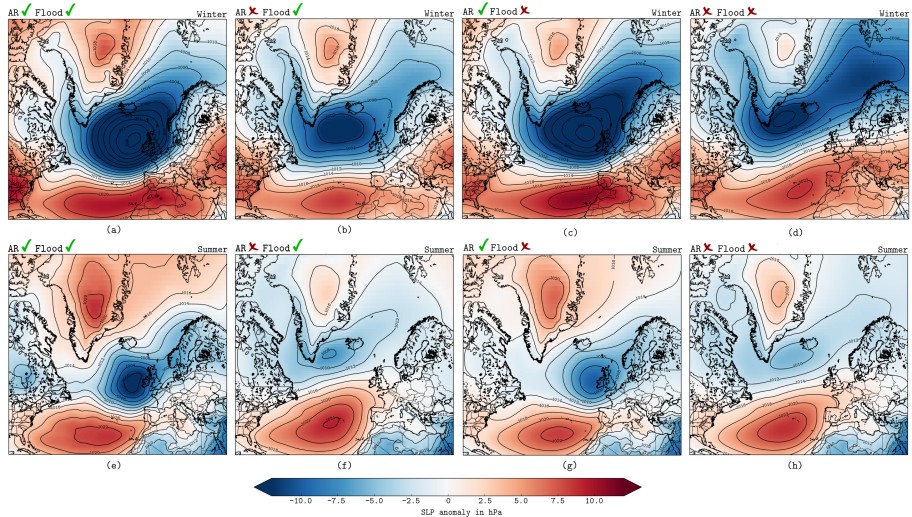

**Figure 6.** SLP composites for all events in the extended winter months depending on whether both an AR and an FE have been detected (a), for FEs only (b), for ARs only (c) for neither (d). (e), (f), (g), (h) respectively represent the same except for the extended summer months. The background field depicts anomalies, and the isolines show mean values. The composites correspond to the period 1979-2010.

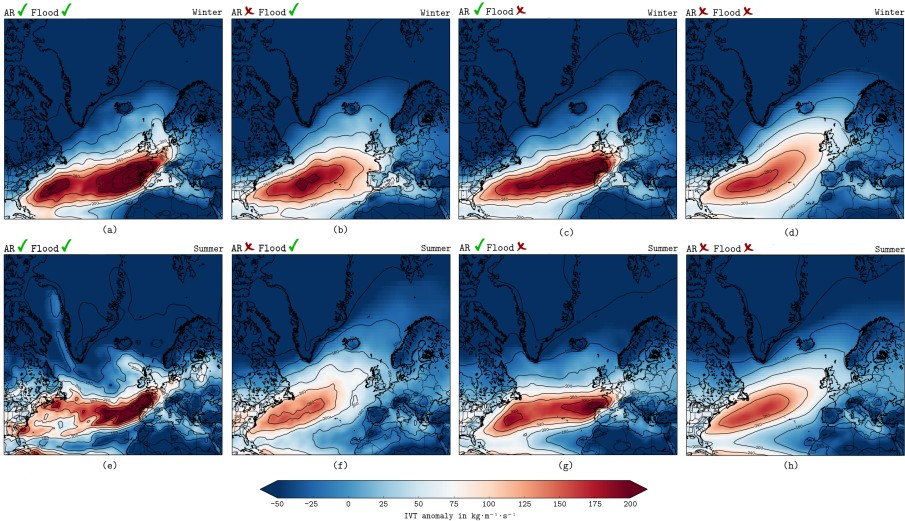

**Figure 7.** Same as Figure 6 but with IVT values.

AR was detected. It is also of note that ~~the floods were associated with disturbances of the IWV fields~~ flooding was associated with disturbances in the IWV fields. These conclusions are similar to those drawn from Figure 6. The results are much less pronounced, ~~since~~due to the fact that the IWV fields are much more stable than the IVT fields. Additionally, the presence of an

AR ~~implies~~suggests a disturbance in the IWV fields that is much more spatially localized than in the case of IVT, and therefore the imprint it leaves on the climatic composite is much lower.

## 4  Conclusions

The flooding episodes in the months between October and March in the coastal areas of Galicia (NW Spain) are associated with WTs of W, NW and C. These WTs are related to inbound baroclinic structures, Atlantic storms and ~~ARS~~ARs ~~to~~on the Galicia coast. Our results support the critical role that ARs play in the intensification of flood episodes, which are present in 70% of the most important FEs in coastal areas, and provide enough moisture to increase the total accumulated rainfall.

~~The link between ARs and FEs is not very evident in coastal areas during the summer months or for the inland basin during any season.~~ The link between ARs and FEs is not so evident in coastal areas throughout the summer months, nor is it evident year round in the inland basin. It is likely that this is due to a more convective nature of precipitation in extreme events far from the coast, and in the extended summer months. It should be noted that most of the FEs in Galicia do not coincide with an AR, in both coastal and inland areas and ~~in~~during both summer and winter months. However, the expected precipitation of an FE is more than doubled if an AR is detected, under any synoptic condition.

The ideas stated in this paper may only be useful to predict FEs ~~together~~along with an operative AR detection algorithm for the Iberian Atlantic Margin. With this aim, we have developed the first operative AR detection system for Europe, which can be accessed at http://meteo.usc.es/ARs. Regarding future work, flood catalogues will allow researchers to perform higher resolution analyses, in order to improve the identification of risk areas, as well as better determine any correlations between floods, ARs, and WTs.

*Acknowledgements.*  The authors acknowledge fruitful discussions with Dr. Swen Brands and MeteoGalicia. Jorge Eiras-Barca is funded by the "Ministerio Español de Economía y Competitividad" and FEDER (EDRF) (CGL2013-45932-R). This work was partially supported by Xunta de Galicia under Project "gts0005 ED431C 2017/64-GRC" Programa de Consolidación e Estruturación de Unidades de Investigacimpetitivas (Grupos de Referencia Competitiva).

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

## Appendix A: Supplementary material

**Table A1.** Information regarding the meteorological stations.

| Estation | Province | Latitude | Longitude | Agency | Type | Region |
|----------|----------|----------|-----------|--------|------|--------|
| Coruña | Coruña | 43°22'1"N | 8°25'9.9"W | AEMET | Manual | COSTA |
| Lourizán | Pontevedra | 42°24'37.8"N | 8°39'46.1"W | MeteoGalicia | Automatic and Manual | COSTA |
| Lugo | Lugo | 43°0'40.0"N | 7°33'17.0"W | AEMET | Manual | SIL |
| Orense | Orense | 42°19'40.0"N | 7°51'37.0"W | AEMET | Manual | SIL |
| Santiago | Coruña | 42°52'12.0"N | 8°33'2.6"W | AEMET | Manual | COSTA |

**Table A2.** Most important FEs listed in descending order of precipitation amount for the COSTA region.

| Event | Region | Day of Max. Prec. | Amount of Prec. (mm) | AR Detection | WT |
|-------|--------|-------------------|----------------------|--------------|-----|
| 13th Oct 1987 - 16th Oct 1987 | COSTA | 14th Oct 1987 | 286.8 | 1 | W |
| 19th Dec 1989 - 21st Dec 1989 | COSTA | 19th Dec 1989 | 253.7 | 1 | SW |
| 07th Sep 1999 - 09th Sep 1999 | COSTA | 08th Mar 1999 | 253.6 | 1 | C |
| 20th Oct 2000 - 10th Jan 2001 | COSTA | 06th Dec 2000 | 202.8 | 1 | SW |
| 02nd Oct 2006 - 25th Oct 2006 | COSTA | 21st Oct 2006 | 189.1 | 1 | SW |
| 11th Nov 2002 - 31st Dec 2002 | COSTA | 12th Nov 2002 | 185.0 | 0 | W |
| 13th Nov 2009 - 29th Dec 2009 | COSTA | 05th Dec 2009 | 182.1 | 1 | SW |
| 21st Feb 2010 - 01st Mar 2010 | COSTA | 21st Feb 2010 | 181.2 | 1 | C |
| 19th Oct 2001 - 23rd Oct 2001 | COSTA | 21st Oct 2001 | 179.9 | 1 | SW |
| 03rd Oct 2010 - 09th Oct 2010 | COSTA | 08th Oct 2010 | 160.0 | 0 | S |
| 21st Dec 1995 - 23rd Jan 1996 | COSTA | 23rd Dec 1995 | 156.9 | 1 | W |
| 05th Jan 1988 - 05th Jan 1988 | COSTA | 05th Jan 1988 | 145.9 | 1 | CW |
| 18th Nov 2006 - 07th Dec 2006 | COSTA | 27th Nov 2006 | 139.2 | 1 | SW |
| 19th Mar 2001 - 24th Mar 2001 | COSTA | 20th Mar 2001 | 105.2 | 1 | CW |
| 24th Apr 2000 - 25th May 2000 | COSTA | 24th Apr 2000 | 103.9 | 1 | CW |
| 28th Oct 2005 - 02nd Nov 2005 | COSTA | 29th Oct 2005 | 103.0 | 1 | SW |
| 12th Jan 2010 - 23rd Jan 2010 | COSTA | 22nd Jan 2010 | 88.2 | 0 | W |
| 10th Nov 1997 - 13th Nov 1997 | COSTA | 10th Nov 1997 | 83.0 | 1 | W |
| 30th Apr 1998 - 01st May 1998 | COSTA | 30th Apr 1998 | 66.0 | 0 | N |
| 09th Jun 2010 - 11th Jun 2010 | COSTA | 10th Jun 2010 | 63.6 | 0 | C |
| 04th Apr 2004 - 08th Sep 2004 | COSTA | 04th Sep 2004 | 62.6 | 0 | ANE |
| 06th Nov 1994 - 06th Nov 1994 | COSTA | 06th Nov 1994 | 41.7 | 0 | CW |
| 23rd Jan 2009 - 25th Jan 2009 | COSTA | 24th Jan 2009 | 40.8 | 1 | W |
| 15th Jun 1988 - 21st Jun 1988 | COSTA | 15th Jun 1988 | 27.9 | 0 | NE |

**Table A3.** Most important FEs listed in descending order of precipitation amount for the SIL region.

| Event | Region | Day of Max. Prec. | Amount of Prec. (mm) | AR Detection | WT |
|---|---|---|---|---|---|
| 14th Oct 1987 - 16th Oct 1987 | SIL | 15th Oct 1987 | 95.4 | 1 | C |
| 31st Dec 1994 - 01st Jan 1995 | SIL | 31st Dec 1994 | 88.7 | 1 | NW |
| 12th Dec 1989 - 24th Dec 1989 | SIL | 16th Dec 1989 | 69.2 | 1 | CW |
| 01st Jan 1994 - 17th Jan 1994 | SIL | 05th Jan 1994 | 68.1 | 1 | W |
| 24th Dec 1995 - 02nd Jan 1996 | SIL | 30th Dec 1995 | 66.6 | 0 | CW |
| 01st Nov 1996 - 30th Nov 1996 | SIL | 22th Nov 1996 | 60.2 | 0 | W |
| 10th Jan 1991 - 12th Jan 1991 | SIL | 10th Jan 1991 | 58.4 | 1 | W |
| 14th Dec 1999 - 16th Dec 1999 | SIL | 14th Dec 1999 | 56.8 | 1 | NW |
| 06th Jan 1996 - 13th Jan 1996 | SIL | 06th Jan 1996 | 50.2 | 0 | CW |
| 15th Jun 1988 - 21st Jun 1988 | SIL | 15th Jun 1988 | 46.4 | 0 | NE |
| 30th Oct 2000 - 31st Mar 2001 | SIL | 21st Nov 2000 | 36.5 | 1 | W |
| 05th Dec 2000 - 13th Dec 2000 | SIL | 07th Dec 2000 | 29.9 | 0 | CW |
| 30th Apr 1998 - 04th May 1998 | SIL | 30th Apr 1998 | 26.6 | 0 | N |
| 01st Nov 2002 - 31st Dec 2002 | SIL | 20th Nov 2002 | 24.7 | 0 | SW |
| 01st Dec 2003 - 31st Dec 2003 | SIL | 09th Dec 2003 | 12.6 | 0 | C |
| 27th Dec 2003 - 27th Dec 2003 | SIL | 27th Dec 2003 | 12.6 | 1 | C |

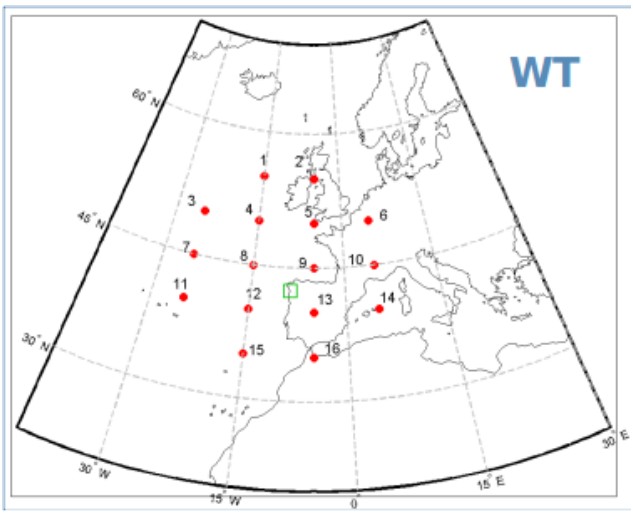

**Figure A1.** Pressure grid points used in the characterization of the WTs.

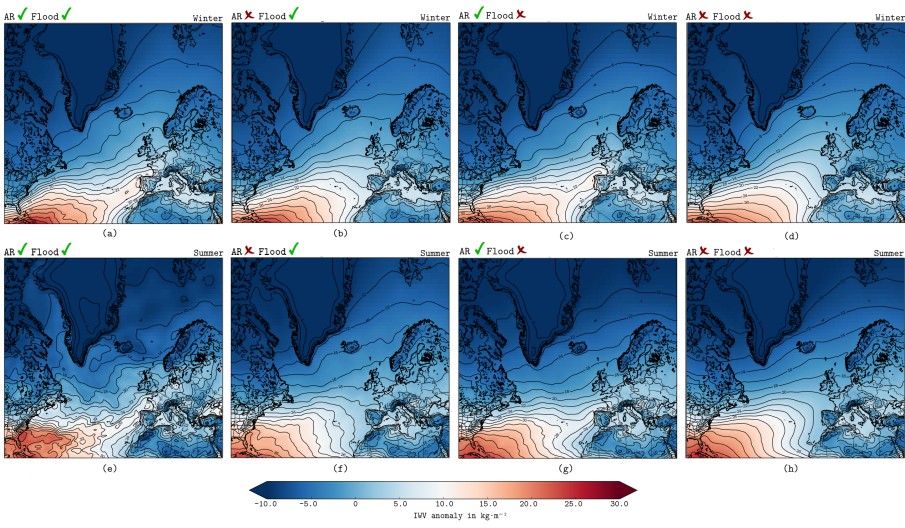

**Figure A2.** Same as Figures 6 and 7, but with IWV values.