# Peer review of "On the relationship between atmospheric rivers, weather types and floods in Galicia (NW Spain)"

_Natural Hazards and Earth System Sciences, 2017_

## Referee Comment (RC1) · Anonymous Referee #1 · 4 Sep 2017

This manuscript analyzes the connection between Atmospheric Rivers and floods under the different weather types obtained in the area of study. The paper is focused in the area of Galicia and another point of interest is the analysis in two separately areas with different characteristics. The manuscript is a good contribution to understand synoptic conditions associated with floods and the influence of AR in severe flooding. It may help to enhance extreme episodes forecasting in Galicia and of interest to emergency services. In my opinion the manuscript is acceptable after minor revisions.

First of all, some details must be added with reference to the data and methods chapter. It would be necessary to give information about the total number of flood events

included in this study. Is any remarkable difference between COSTA and SIL number of flood events? Furthermore, I think this database contains flood events due to some causes as river flooding, drainage problems, flood events, etc. For a better understanding, it would be of interest mention if the manuscript takes into account all of them or it is focused on a selection.

On the other hand, there is not a lot of information about the data used for the synoptic classification; is it supposed that SLP values are provided by NCAR for the same period? There is the reference to the paper carried out by Trigo et al., 2000; are the same 16 grid points used in both studies? The rules to define different types of circulation are supposed to be the same established by Trigo et al., 2000, can you confirm it? Finally, I suggest including a table in this chapter with the WTs associated to the extended winter and summer months with the description of each class of WTs.

In section 3 the frequency of each WTs is presented. Are the results in concordance with other WTs studies carried out in the same area for precipitation episodes?

In the introduction section is stated as one of the scopes that this study may be useful to properly understand and predict the damages caused by FEs. The flood events database includes damage information for each event (injured people, damages in property, etc). However, in the results section this information is not presented. Have you analyzed it? If a damage analysis has not been carried out, it would be necessary to eliminate any damage reference or replace by the prediction of precipitation amounts (presented in section 3.1).

In the conclusions you sum up the most remarkable results obtained. In my opinion it is necessary to give some remarks about future work and how to integrate this information as a useful tool in emergency warnings.

Finally, as specific suggestions on page number 6 lines 5-6 there is an internal comment, figure 5 description is not correct (precipitation ratio instead of frequency of occurrence), description of Table A1 an A2 probably is in terms of amount of precipitation

instead of damage and it would be better another English revision because some sentences are difficult to understand.

---

## Referee Comment (RC2) · Anonymous Referee #2 · 29 Sep 2017

Below is my review of the manuscript entitled "On the Relationship Between Atmospheric Rivers, Weather Types and Floods in Galicia". The article talks about the significant contribution of the atmospheric rivers to major floods in northwestern Spain. This is such an interesting and relatively new climate research. The results may improve the flood emergency plans in Galicia. The manuscript contains a high scientific quality and methods are relevant to obtain the results. On the other hand, some discussion of the results is missing. Furthermore, the writing might be improved.

MAJOR COMMENTS

1) Writing needs improving. Many short paragraphs should be joined with other paragraphs. Other details are listed in specific comments. 2) Discussion is missing in the article. Only L6-7P7 appear to be discussed. Please include other previous works when writing the discussion. I would propose to name section 3 as "Results and discussion". 3) Several figures must be improved: Figure 2: What do the different colours mean? Please add a legend to the map. Furthermore, an N arrow and a scale bar are missing. Figure 3: It shows 14 WTs, but in the text you talk about 9 WTs in winter and 12 WTs summer. It is important to specify in the text and in Figure 3 which ones are for winter and which are for summer. Figures 4 and 5: The scale units in y axis have to be the same in all 4 plots for comparison. Figure 4 and 6: Please consider writing (a), (b), etc. at the beginning of the text of each subfigure. Figure 5: Precipitation ratio is not mentioned in the caption. 4) Methodology section: Some description about how precipitation ratio is calculated would be welcome. Furthermore, any comment about computing anomaly maps could be also included.

SPECIFIC COMMENTS

L4P1: Please add "northwestern" to this sentence: "…and floods in the northwestern Spanish region…" L16P1: Please rewrite as "3,700 km". L1P2: Please replace "integrated vapor transport fields (IVT)" with "IVT fields". L3-4P2: These lines should be included in the next paragraph. It cannot be an only paragraph of two lines. L9-10P2: The following sentence "An example of a well-defined……, can be found in Figure 1" should be shifted to L2P2, after "surrounding areas". L13P2: Please replace "fall" with "autumn". L1P3: Please add ", above all in southern Galicia," after "variability in precipitation". L3P3: Please replace "SCA" with "SCAND". L4P3: Please exchange the positions between "Bueh and Nakamura (2007)" and "Lorenzo et al. (2006)". L5P3: Please rewrite as follows "Extreme precipitation and anomalous winds are the most frequent…". FEs is somehow already derived from heavy precipitation and strong winds over the coast. L7P3: "…by no means abnormal" sounds weird. Please review it. L11P3: Please replace "WT" with "WTs". L14P3: Please replace "flood episodes" with "FEs". L5-6P4: These two lines should be added to the preceding paragraph.

L9P4: Please specify what is "Interior". This is not a family name. L20-21P4: Please join it with the following paragraph. L22P4: Please rewrite as "...developed in Trigo et al. (2000), which..." L28P4: Please replace "nine" with "9". L3-4P5: These two lines should be added to the preceding paragraph. L4P5: Please replace "The" with "the". L9P5: Please replace "A3" with "A1". L12-13P5: These two sentences must be deleted. This content corresponds to the caption of Figure 4, where you already explained the colours of the bars. L5P6: Please rewrite this sentence. The meaning is not clear. L5-6P6: Please delete "meaning unclear please clarify". L10P6: Please replace "floods" with "FEs". L2P7: Please replace "Table 1" with "Table A2". L4P7: Please replace "case" with " region (Table A3)". L9P7: Please delete "In other words". L12-13P7: Please rewrite as follows "Especially noteworthy is the case of the ANW type for the SIL and COSTA regions and NW for SIL region, where the expected winter rainfall is five times larger than..." L14P7: Please rewrite "for types SW, W and C for the winter..." L15-16P7: Please rewrite the sentence as follows (for example): "In the summer months, the most outstanding cases are W type in SIL region and AN type in COSTA region". When you write "the same occurs..." is difficult to associate with the "five times larger...". L16-18P7: I do not follow the meaning of the "exception" you mention. I do not understand what you mean with the last sentence when I look at Figure 4. Please review these last two sentences. L34P7: Please rewrite as follows: "detection of an AR in winter months is contingent on the..." L2P9: Please replace "SW" with "NW". L3P9: Please replace "atmospheric rivers" with "atmospheric rivers (ARs)". Equation 2: It needs further explanation of some variables. Figure 1: Please rewrite the first sentence of the caption as follows "Example of an atmospheric river (AR) landfalling in Galicia (NW Spain) on 5th January 2001". Figure 4: Please replace "(ONDJFM" with "(ONDJFM)". Figure 5: Please rewrite as follows: "Same as Figure 4 but with precipitation ratio. Table A1: Please replace "A1" with "A2". Please replace "07/09/99" with "03/07/99". Table A2: Please replace "A2" with "A3". Missing data in the last event. Old tables A1 and A2: Dates must follow English style mm/dd/yy. Some events take more than a month, is it correct? For instance, 4th, 6th and 7th events in

old Table A1; 14th in old Table A2. I do not think that a rainfall event can take longer than a week. If this is the case, please justify what you consider for a precipitation event in the Data and Methods section. Table A3: Please replace "A3" with "A1". Geographic coordinates are in decimal degrees in Santiago, please replace it with degrees, minutes and seconds. Please replace "Coast" and "Miño-Sil" with "COSTA" and "SIL", respectively, in the last column. Figure A1: Please rewrite as follows: "...of the weather types (WTs)".
* * *

---

## Author Comment (AC1) · 22 Nov 2017

We would like to honestly thank the reviewer for the positive review and his/her valuable comments. All of them are going to be taken into account for the final version of the manuscript.

Please, find below the list with the one-to-one reviewer's comments addressed.

**First of all, some details must be added with reference to the data and methods chapter. It would be necessary to give information about the total number of flood events included in this study. Is any remarkable difference between COSTA and SIL number of flood events? Furthermore, I think this database contains flood events due to some causes as river flooding, drainage problems, flood events, etc. For a better understanding, it would be of interest mention if the manuscript takes into account all of them or it is focused on a selection.**

We agree with the reviewer in the fact that it is necessary to include a discussion on the total number of flood events analyzed in this paper. The following paragraph is going to be included in the final version of the manuscript:

*"A total number of 754 AR events have been detected for the Galician coast throughout the period 1978-2010. In the same period, 357 and 548 flood events have been registered for MIÑO-SIL and COSTA REGION respectively. The seasonal distribution of flood events is stated in Table 1".*

| Season | MIÑO-SIL Region | COSTA Region |
|--------|-----------------|--------------|
| DJF | 206 | 190 |
| MAM | 45 | 102 |
| JJA | 9 | 102 |
| SON | 97 | 154 |

*Table 1: Total number of flood events registered in Galicia throughout the period 1978-2010.*

**On the other hand, there is not a lot of information about the data used for the synoptic classification; is it supposed that SLP values are provided by NCAR for the same period? There is the reference to the paper carried out by Trigo et al., 2000; are the same 16 grid points used in both studies? The rules to define different types of circulation are supposed to be the same established by Trigo et al., 2000, can you confirm it?**

For the index calculations we applied the equations, data and procedured outlined in Lorenzo et al. (2008). This means:

- 16 grid points and procedure obtained from Trigo et al. (2000). Our points are dispacel 5 deg. northward in order to cover the region of Galicia instead of Portugal.

- SLP data obtained from NCAR 2.5x2.5 reanalysis data.

This point is going to be further clarified in the latest version of the manuscript in this way:

P4L25 *"(…)(SLP) values collected for the 16 grid points shown in the supplementary material Figure A1 (Lorenzo et al., 2008).*

*For the index calculations, we applied the equations outlined in Lorenzo et al. (2008) to NCAR reanalysis 2.5deg · 2.5deg data."*

**Finally, I suggest including a table in this chapter with the WTs associated to the extended winter and summer months with the description of each class of WTs.**

The table suggested by the reviewer is going to be added to the "Section 2: Data and Methods" as a complement to Figure 3. The description has been obtained from :

Ramos, A.M. : *Improving circulation weather type classifications using a 3D framework: relationship with climate variability and projections for future climates*. PhD Thesis, UVIGO, 2012.

| WT | Season | Brief description |
|----|--------|-------------------|
| NE | S | Days characterized by an extended high pressure settled over the west of Ireland and low pressure in the Mediterranean Sea. |
| E | S | Synoptic situations characterized by high pressure over the British Isles and low pressure dominating in North Africa. |
| SE | W | Low pressure extending towards Madeira and high pressure over Northern Europe. |
| S | S,W | Pressure over the British Isles and low pressure established in the North Atlantic (Azores region). |
| SW | W | Low-pressure system to the west of Ireland with a large anticyclone over the Mediterranean region. |
| W | S,W | Lowpressure system over the North Atlantic, with a high-pressure system over the Azores. |
| NW | S,W | Low-pressure system over the British Islands nda n anticyclone system located over Azores. |
| N | S | Presence of the Azores high pressure near the Azores Islands and a low pressure over southern Europe and the Mediterranean basin. |
| C | S,W | Lowpressure centre over the NW Iberian Peninsula. |
| A | S,W | Extended highpressure centre between the Iberian Peninsula and the Azores Islands. |

**In section 3 the frequency of each WTs is presented. Are the results in concordance with other WTs studies carried out in the same area for precipitation episodes?**

This manuscript concludes that flooding episodes in the months between October and March in the coastal areas of Galicia are associated with WTs of W, SW and C. To the best of our knowlege, all the WTs studies carried out including Galicia and/or Portugal are in agreement with this conclusion (Trigo and Dacamara, 2000; Lorenzo et al. 2008; Cortesi et al. 2013). WTs W, SW and C are identified as the most important precursos for precipitation in all of them.

We agree with the reviewer in the fact that this should be discussed in the manuscript. Thus, this idea is going to be included in the "Results" section in the final version of the manuscript as follows:

The section is going to include the paragraph:

*"Previous works analyzing precipitation from the methodology of WTs conclude that most of the yearly and winter precipitation is associated with WTs C, SW and W (e.g. Trigo and DaCamara 2000; Lorenzo et al., 2008; Cortesi et al., 2014). Our results are in complete agreement with those obtained by the cited studies adding the idea that ARs are responsible for most of this precipitation."*

Cortesi, N., Gonzalez-Hidalgo, J. C., Trigo, R. M., & Ramos, A. M. (2014). Weather types and spatial variability of precipitation in the Iberian Peninsula. *International Journal of Climatology*, *34*(8), 2661-2677.

Lorenzo, M. N., Taboada, J. J., & Gimeno, L. (2008). Links between circulation weather types and teleconnection patterns and their influence on precipitation patterns in Galicia (NW Spain). *International Journal of Climatology*, *28*(11), 1493-1505.

Trigo, R. M., & DaCAMARA, C. C. (2000). Circulation weather types and their influence on the precipitation regime in Portugal. *International Journal of Climatology*, *20*(13), 1559-1581.

**In the introduction section is stated as one of the scopes that this study may be useful to properly understand and predict the damages caused by FEs. The flood events database includes damage information for each event (injured people, damages in property, etc). However, in the results section this information is not presented. Have you analyzed it? If a damage analysis has not been carried out, it would be necessary to eliminate any damage reference or replace by the prediction of precipitation amounts (presented in section 3.1).**

Certainly, this manuscript does not foresee "damages" themselves. However, we kindly ask the referee to take into account that flooding episodes have been obtained from an Emergency System database. Thus, all of them were associated with damages one way or another. However, following the advice of the reviewer, this point is going to be clarified in the final version of the manuscript as follows:

*"Next, we show that this study may be useful to properly understand and predict the damages caused by FEs."*

will be replaced by

*"Next, we show that this study may be useful to properly understand and predict FEs."*

**In the conclusions you sum up the most remarkable results obtained. In my opinion it is necessary to give some remarks about future work and how to integrate this information as a useful tool in emergency warnings.**

Following the advice of the reviewer, we are going to add this paragraph to the conclusions:

*"The ideas stated in this paper may only be useful to predict flood events together with an operative AR detection algorithm for the Iberian Atlantic Margin. With this aim, we have developed the first operative AR detection system for Europe, which can be accessed at [http://meteo.usc.es/ARs](http://meteo.usc.es/ARs). Regarding future work, flood catalogues will allow to perform higher resolution analyses, in order to improve the identification of risk areas, as well as the correlation between floods, ARs and WTs."*

**Finally, as specific suggestions on page number 6 lines 5-6 there is an internal comment.**

The internal comment is going to be removed in the latest version of the manuscript.

**Figure 5 description is not correct (precipitation ratio instead of frequency of occurrence).**

This is a typo that will be fixed in the final version of the manuscript.

**Description of Table A1 an A2 probably is in terms of amount of precipitation instead of damage.**

Tables A1 and A2 show the most important precipitation in terms of damage, and they are listed in order of observed precipitation. *"Listed in order of measured precipitation"* is going to be added to the caption of the tables.

**It would be better another English revision because some sentences are difficult to understand.**

Following the reviewer's suggestion, we are going to order a new native english revision. Changes regarding specifically the language are going to be highlighted in blue instead of red in the final version of the manuscript

---

## Author Comment (AC2) · 22 Nov 2017

We would like to thank the reviewer for his/her thourough review of this manuscript. It will be very helpful to improve the final version of the paper.

**MAJOR COMMENTS**

**1) Writing needs improving. Many short paragraphs should be joined with other paragraphs. Other details are listed in specific comments.**

Following the advice of the reviewer, we are not going to submit the final version of the manuscript without a previous checking from a native reader.

**2) Discussion is missing in the article. Only appear to be discussed. Please include other previous works when writing the discussion. I would propose to name section 3 as "Results and discussion".**

Section 3 is going to be renamed as "Section 3 : Results and discussion."

The section is going to include the paragraph:

*"Previous works analyzing precipitation from the methodology of the WTs conclude that most of the yearly and winter precipitation is associated to WTs C, SW and W (e.g. Trigo and DaCamara 2000; Lorenzo et al., 2008; Cortesi et al., 2014). Our results are in complete agreement with those obtained by the cited studies adding the idea that ARs are responsible for most of this precipitation."*

Cortesi, N., Gonzalez-Hidalgo, J. C., Trigo, R. M., & Ramos, A. M. (2014). Weather types and spatial variability of precipitation in the Iberian Peninsula. *International Journal of Climatology*, *34*(8), 2661-2677.

Lorenzo, M. N., Taboada, J. J., & Gimeno, L. (2008). Links between circulation weather types and teleconnection patterns and their influence on precipitation patterns in Galicia (NW Spain). *International Journal of Climatology*, *28*(11), 1493-1505.

Trigo, R. M., & DaCAMARA, C. C. (2000). Circulation weather types and their influence on the precipitation regime in Portugal. *International Journal of Climatology*, *20*(13), 1559-1581.

**3) Several figures must be improved: Figure 2: What do the different colours mean? Please add a legend to the map. Furthermore, an N arrow and a scale bar are missing.**

The different colors in Figure 2 were a political division, which is going to be removed in the final version of the manuscript. The arrow with the scale is going to be added in the final version of the manuscript.

[Figure]

2001-01-05 12h

[Figure]

IVT=400 kg·m⁻¹·s⁻¹     Water Vapor in kg·m⁻²

**Figure 3: It shows 14 WTs, but in the text you talk about 9 WTs in winter and 12 WTs summer. It is important to specify in the text and in Figure 3 which ones are for winter and which are for summer.**

Following the advice of another reviewer, we are going to add a table where this issue is going to be addressed.

| WT | Season | Brief description |
|----|--------|-------------------|
| NE | S | Days characterized by an extended high pressure settled over the west of Ireland and low pressure in the Mediterranean Sea. |
| E | S | Synoptic situations characterized by high pressure over the British Isles and low pressure dominating in North Africa. |
| SE | W | Low pressure extending towards Madeira and high pressure over Northern Europe. |
| S | S,W | Pressure over the British Isles and low pressure established in the North Atlantic (Azores region). |
| SW | W | Low-pressure system to the west of Ireland with a large anticyclone over the Mediterranean region. |
| W | S,W | Lowpressure system over the North Atlantic, with a high-pressure system over the Azores. |
| NW | S,W | Low-pressure system over the British Islands nda n anticyclone system located over Azores. |
| N | S | Presence of the Azores high pressure near the Azores Islands and a low pressure over southern Europe and the Mediterranean basin. |
| C | S,W | Lowpressure centre over the NW Iberian Peninsula. |
| A | S,W | Extended highpressure centre between the Iberian Peninsula and the Azores Islands. |

**Figures 4 and 5: The scale units in y axis have to be the same in all 4 plots for comparison.**

In the final version of the manuscript, the scale units in the y axis are going to be the same in all four plots for comparison.

**Figure 4 and 6: Please consider writing (a), (b), etc. at the beginning of the text of each subfigure.**

The reviewer´s advice will be followed.

**Figure 5: Precipitation ratio is not mentioned in the caption.**

"Frequency of occurrence" is going to be replaced by *"precipitation ratio"* in the caption of the final version of the manuscript.

**4) Methodology section: Some description about how precipitation ratio is calculated would be welcome. Furthermore, any comment about computing anomaly maps could be also included.**

The precipitation ratio is calculated as follows:

First, the mean precipitation is calculated for each WT, cumputing the accumulated precipitation during the flooding days in coincidence with the WT under consideration.

Secondly, we have calculated the mean precipitation in flood events with (and without) AR event.

The ratio is the result of the division of the latter by the former.

Even when this procedure is explained in P7L 8-12, this description is going to be added to the final version of the manuscript.

**SPECIFIC COMMENTS**

**L4P1: Please add "northwestern" to this sentence: "and floods in the northwestern Spanish region."**

The sentence is going to be rewritten as proposed.

**L16P1: Please rewrite as "3,700 km".**

The amount is going to be rewritten, but we would rather to do it as "3,700 km".

**L1P2: Please replace "integrated vapor transport fields (IVT)" with "IVT fields".**

"integrated vapor transport fields (IVT) is going to be replaced with "IVT fields".

**L3-4P2: These lines should be included in the next paragraph. It cannot be an only paragraph of two lines.**

We are going to unify both paragraphs in the final version of the manuscript.

**L9-10P2: The following sentence "An example of a well-defined .....:, can be found in Figure 1" should be shifted to L2P2, after "surrounding areas".**

This is going to be done in the final version of the manuscript.

**L13P2: Please replace "fall" with "autumn".**

"Fall" is going to be replaced by "autumn" in the final version of the manuscript.

**L1P3: Please add ", above all in southern Galicia," after "variability in precipitation".**

The proposed sentence is going to be added in the final version of the manuscript.

**L3P3: Please replace "SCA" with "SCAND".**

"SCA" is going to be replaced by "SCAND" in the final version of the manuscript.

**L4P3: Please exchange the positions between "Bueh and Nakamura (2007)" and "Lorenzo et al. (2006)".**

The positions are going to be exchanged in the final version of the manuscript.

**L5P3: Please rewrite as follows "Extreme precipitation and anomalous winds are the most frequent**

**...". FEs is somehow already derived from heavy precipitation and strong winds over the coast.**

The sentence is going to read as proposed, in the final version of the manuscript.

**L7P3: "by no means abnormal" sounds weird. Please review it.**

*"by no means abnormal"* is going to be replaced by *"are common"* in the final version of the manuscript.

**L11P3: Please replace "WT" with "WTs".**

"WT" is going to be replaced by "WTs".

**L14P3: Please replace "flood episodes" with "FEs".**

*"flood episodes"* is going to be replaced by *"FEs".*

**L5-6P4: These two lines should be added to the preceding paragraph.**

Both lines are going to be added to the preceding paragraph in the final version of the manuscript.

**L9P4: Please specify what is "Interior". This is not a family name.**

This is just a citation to:

Interior, M.: Catálogo Nacional de Inundaciones Históricas, Dirección General de Protección Civil y Emergencias. Ministerio del Interior español, 2014.

**L20-21P4: Please join it with the following paragraph.**

This paragraph is going to be joined with the following paragraph.

**L22P4: Please rewrite as "(...) developed in Trigo et al. (2000), which (...)".**

The sentence is going to be rewritten as proposed.

**L28P4: Please replace "nine" with "9".**

"Nine" is going to be replaced with "9".

**L3-4P5: These two lines should be added to the preceding paragraph.**

Both lines are going to be added to the preceding paragraph.

**L4P5: Please replace "The" with "the".**

"The" is going to be replaced with "the".

**L9P5: Please replace "A3" with "A1".**

The order of the tables is going to be fixed in the final version of the manuscript.

**L12-13P5: These two sentences must be deleted. This content corresponds to the caption of Figure 4, where you already explained the colours of the bars.**

The two sentences are going to be deleted.

**L5P6: Please rewrite this sentence. The meaning is not clear.**

The sentence is going to be rewritten as follows:

*"Especially remarkable is the case of the SW, which registers as many floods associated with AR events as floods not associated with AR events."*

**L5-6P6: Please delete "meaning unclear please clarify".**

This is going to be removed.

**L10P6: Please replace "floods" with "FEs".**

"floods" is going to replaced with its acronym.

**L2P7: Please replace "Table 1" with "Table A2".**

The order of the tables is going to be fixed in the final version of the manuscript.

**L4P7: Please replace "case" with " region (Table A3)".**

This change is going to be made in the final version of the manuscript.

**L9P7: Please delete "In other words".**

"In other words" is going to be deleted.

**L12-13P7: Please rewrite as follows "Especially noteworthy is the case of the ANW type for the SIL and COSTA regions and NW for SIL region, where the expected winter rainfall is five times larger than…"**

We are going to rewrite the text as proposed.

**L14P7: Please rewrite "for types SW, W and C for the winter".**

We are going to rewrite as proposed.

**L15-16P7: Please rewrite the sentence as follows (for example): "In the summer months, the most outstanding cases are W type in SIL region and AN type in COSTA region". When you write "the same occurs (…)" is difficult to associate with the "five times larger.**

We are going to rewrite as proposed.

**L16-18P7: I do not follow the meaning of the "exception" you mention. I do not understand what you mean with the last sentence when I look at Figure 4. Please review these last two sentences.**

Me meant to say that SW weather types with the arrival of oceanic fronts are not common in summer months in Galicia, when N/NE are the norm. For the sake of clarity, the full sentences are going to be rewritten as follows:

*"Even when SW is not a predominant weather type in summer months, the occurrence of this WT in this season represents the few fronts with ARs arriving on the Galician coast. It is uncommon to observe AR precipitation from WTs other than W or SW in summer months."*

**L34P7: Please rewrite as follows: "detection of an AR in winter months is contingent on the (…)"**

We are going to rewrite as proposed.

**L2P9: Please replace "SW" with "NW".**

We have to disagree here, we have rechecked Figures 5.c and 4.c, and we find "SW" to be correct.

**L3P9: Please replace "atmospheric rivers" with "atmospheric rivers (ARs)".**

We are going to rewrite as proposed.

**Equation 2: It needs further explanation of some variables.**

"Equations 1 and 2 represent the methodology for the integration of the IWV and IVT fields, respectively, where q is the specific humidity, g is the gravitational force, and the integration covers the whole troposphere."

is going to be replaced by:

*"Equations 1 and 2 represent the methodology for the integration of the IWV and IVT fields, respectively, where q is the specific humidity, g is the gravitational force, **u** is the horizontal wind field and the integration covers the whole troposphere from the first pressure level (Po) to the top (Pf)."*

**Figure 1: Please rewrite the first sentence of the caption as follows "Example of an atmospheric river (AR) landfalling in Galicia (NW Spain) on 5th January 2001".**

We are going to rewrite as proposed.

**Figure 4: Please replace "(ONDJFM" with "(ONDJFM)".**

We are going to rewrite as proposed.

**Figure 5: Please rewrite as follows: "Same as Figure 4 but with precipitation ratio.**

We are going to rewrite as proposed.

**Table A1: Please replace "A1" with "A2".**

The order of the tables is going to be fixed in the final version of the manuscript.

**Please replace "07/09/99" with "03/07/99". Table A2: Please replace "A2" with "A3".**

The order of the tables is going to be fixed in the final version of the manuscript.

**Missing data in the last event. Old tables A1 and A2: Dates must follow English style mm/dd/yy. Some events take more than a month, is it correct? For instance, 4th, 6th and 7th events in old Table A1; 14th in old Table A2. I do not think that a rainfall event can take longer than a week. If this is the case, please justify what you consider for a precipitation event in the Data and Methods section.**

We are going to follow the advices regarding the english style of dates. Regarding the length of the flood events, the reviewer should consider that the authors took the flood events directly from the emergency system service database. The final version of the manuscript is going to include a discussion about this in Data and Methods section.

**Table A3: Please replace "A3" with "A1". Geographic coordinates are in decimal degrees in Santiago, please replace it with degrees, minutes and seconds. Please replace "Coast" and**

**"Miño-Sil" with "COSTA" and "SIL", respectively, in the last column. Figure A1: Please rewrite as follows: "(…) of the weather types (WTs)".**

We are going to rewrite as proposed.

---

## Author Response (AR1)

Dear editor María del Carmen Llasat,

Firs of all, we would like to thank you for your review process which helped to improve the article substantially. We have followed all the instructions given by you and the reviewers. Particularly:

- The entire text has been revised by an official science related translator (International Science Editing, Bay K, Unit 11a, Shannon, Ireland). You can find the language-related changes highlighted in blue color.

- We have addressed all the reviewer's comments. You can find these changes highlighted in color red. The most important changes that  we have made in this latest version of the manuscript are:

  ◆ We have added two important tables; the first table is related to the number of events and the second tables gives a brief description of each weather type considered in the analysis.

  ◆ We have updated Figures 4 and 5, so that all the y-axis have the same limits to facilitate the comparison between them.

  ◆ We have extended the discussion about the previous works published on this matter.

  ◆ We have adapted all the dates to the English writing style.

We hope this changes will meet you expectations.
Kind Regards,

Jorge Eiras-Barca

---

## Referee Report (RR1)

The authors have taken into account the suggestions received during the discussion process and some data have been introduced that helps to a better understanding. In my opinion, information on the number of flood events and their distribution was necessary to justify the importance on the results obtained because there was no other quantitative reference. Thus, Table number 1 and other corrections as the references introduced in the discussion, give more value to the study. However, there are still corrections to be introduced, so this article could be accepted with minor corrections.

First of all, in Section Data and Methods it seems to be a contradiction between the period included with the corrections and the period stated in lane number two. I think that both periods should be 1978 – 2010 (in the text appears 1979).

On the other hand, not all weather types on Figures 4 and 5 appear on Figure 3, and vice versa. That's the case of ASW winter WT and ANE summer WT for example. You must check the 14 WT considered in your analysis with the WT presented in the following figures and sections. If there is a mistake in some Figures, they should be modified.

Finally, you should review the text because there are some mistakes. For example, in Figure number 1 "AR" is missing in the text and on Table number 2 we can find "low pressure", "low-pressure" and "lowpressure".

---

## Author Response (AR2)

Dear editor Maria Carmen Llasat,

We would like to thank your commitment throughout the revision process. Following your instructions, we have been throughout all the reviewer's advices. Most of them helped to improve the article substantially. These changes are highlighted in color red in the final version of the manuscript.

Additionally, we have resent the entire manuscript to a professional native writer, who helped to increase the formality and the clarity of the text. These changes are highlighted in color blue.

We hope that you can find this version of the manuscript acceptable for publishing.

Kind regards,
Jorge Eiras-Barca.

---

## Author Response (AR3)

Dear Prof. Llasat,

We would like to thank you for your positive assessment of version number 5 of our manuscript. Please, find below the replies to your latest comments.

**The first one refers to line 18 where you say that 357 and 548 flood events have been recorded for the SIL and COSTA region in Galicia, respectively, during the period 1979-2010. These figures are so much high. If we understand flood event as a meteorological or hydrological event that has given place to one or more floods, they are unrealistic. Perhaps you refer to the number of claims made to insurance companies as a consequence of floods, or the addition of the total number of municipalities affected by each event. The catalogue made by Civil Protection only includes major events. Then it is not clear what do you understand as flood event and which is the source. I suggest you to include in the paper what do you understand as flood event, and to enlarge table 1 with two columns including the number of meteorological/hydrological events or/and the total number of days affected by FE.**

We agree with the editor in that the paragraph is somewhat confusing and should be rewritten. With these figures we are talking about the total number of days encompassed by all considered flood events. In the period 1979-2010 there are 24 events (i.e. flooding episodes or floods) for the COSTA basin and 15 events for the SIL basin (Tables A2 and A3), but it must be taken into account that some flood episodes in each basin persist for several days, hence the higher numbers mentioned erroneously as flood events.

Therefore we propose to rewrite the paragraph:

"A total number of 754 AR events have been detected on the Galician coast throughout years 1979-2010. In the same period, 357 and 548 flood events were registered for the SIL and COSTA regions respectively . The seasonal distribution of FEs is provided in Table 1."

as

*"The total number of days where an AR has been detected on the Galician coast during years 1979-2010 is 754. We note that the same AR event can extend for more than one day. In the same period, 24 and 15 flood events were registered for the SIL and COSTA regions respectively. Since each FE can also last for several days (see Table 1), the total number of days encompassed by all considered FEs is 357 for the SIL and 548 for the COSTA regions".*

Additionally, "Table 1. Total number of FEs registered in Galicia throughout the period 1979-2010." has been rewritten as *"Table 1. Total number of days included in any of the flood events in Galicia throughout the period 1979-2010."*

Finally, for the sake of clarity, in tables A2 and A3, where used to read "Most important FEs..." now reads *"FEs included in the analysis, listed in ... (..)"*

**Although I am reluctant to include in my reviewing or editing reports my own references, in this case I consider it is unavoidable to propose you to complete your discussion with the paper "Gilabert, J. and M.C. Llasat, 2017. Circulation weather types associated with extreme flood events in Northwestern Mediterranean. Int. J. Climatol. (2017) Published online in Wiley Online Library (wileyonlinelibrary.com) DOI: 10.1002/joc.5301)". In this paper WT obtained with your same methodology have been applied to the analysis of flood events in the NE part of the Iberian Peninsula. I think the comparison with the results obtained in this other paper for a near Mediterranean region placed at the same latitude will provide a major robustness to your paper.**

We have carefully read the paper the editor is proposing for citation. Please, keep in mind that even when both studies use an adaptation of the Lamb methodology for the WT classification, the Flood databases are completely different (as they are the criteria to identify the events). Additionally, the regions analized differ strongly in terms of climate. While Galician floods are mostly associated to cyclonic WTs (Atlatic winds), which usually lead to the landfall of atmospheric rivers; Catalonian floods are mostly produced by Mediterranean air masses from the SE or advective flows from the N (jointly with a key role played by convection). Additionally, it is important to note that flood events are longer lasting in Galicia than in Catalonia. Despite this, Catalonian events tend to be much more devastating that the regular Galician events. Finally, we think that it is important to note that our manuscript has not been based in Gilabert and Llasat (2017), since both reviewing processes occurred simultaneously.

For all these reasons, we think that the citation to Gilabert and Llasat (2017) could fit in the Introduction, together with other previous works which have applied the WT methodology.